# Two-Stream Convolutional Neural Networks for Breathing Pattern Classification: Real-Time Monitoring of Respiratory Disease Patients

**DOI:** 10.3390/bioengineering11070709

**Published:** 2024-07-12

**Authors:** Jinho Park, Thien Nguyen, Soongho Park, Brian Hill, Babak Shadgan, Amir Gandjbakhche

**Affiliations:** 1Eunice Kennedy Shriver National Institute of Child Health and Human Development, National Institutes of Health, 49 Convent Dr., Bethesda, MD 20894, USA; jinho.park@nih.gov (J.P.); thien.nguyen4@nih.gov (T.N.); soongho.park@nih.gov (S.P.); brian.hill@nih.gov (B.H.); 2National Heart, Lung and Blood Institute, National Institutes of Health, 10 Center Dr., Bethesda, MD 20892, USA; 3Implantable Biosensing Laboratory, International Collaboration on Repair Discoveries, Vancouver, BC V5Z 1M9, Canada; babak.shadgan@ubc.ca; 4Department of Pathology & Laboratory Medicine, University of British Columbia, Vancouver, BC V6T 1Z7, Canada

**Keywords:** COVID-19, deep learning, convolutional neural network, respiratory disease, NIRS, wearable device

## Abstract

A two-stream convolutional neural network (TCNN) for breathing pattern classification has been devised for the continuous monitoring of patients with infectious respiratory diseases. The TCNN consists of a convolutional neural network (CNN)-based autoencoder and classifier. The encoder of the autoencoder generates deep compressed feature maps, which contain the most important information constituting data. These maps are concatenated with feature maps generated by the classifier to classify breathing patterns. The TCNN, single-stream CNN (SCNN), and state-of-the-art classification models were applied to classify four breathing patterns: normal, slow, rapid, and breath holding. The input data consisted of chest tissue hemodynamic responses measured using a wearable near-infrared spectroscopy device on 14 healthy adult participants. Among the classification models evaluated, random forest had the lowest classification accuracy at 88.49%, while the TCNN achieved the highest classification accuracy at 94.63%. In addition, the proposed TCNN performed 2.6% better in terms of classification accuracy than an SCNN (without an autoencoder). Moreover, the TCNN mitigates the issue of declining learning performance with increasing network depth, as observed in the SCNN model. These results prove the robustness of the TCNN in classifying breathing patterns despite using a significantly smaller number of parameters and computations compared to state-of-the-art classification models.

## 1. Introduction

In 2019, the respiratory illness Coronavirus Disease 2019 (COVID-19) caused by SARS-CoV-2 started spreading globally at an unprecedented rate. The virus has spread to over 228 countries and, as of May 2024, has affected approximately 775 million people, resulting in over 7.05 million deaths [1]. Common symptoms of COVID-19 include cough, breathing complications, wheezing, and tachypnea, which often accompany respiratory distress, fever, headache, fatigue, hypoxia, and lethargy [2,3,4,5,6]. Various methods have been used to monitor infected patients, such as CT scans [7,8,9], X-rays [7,10], thermal cameras [11], depth cameras [4], ultrasound technology [12], and radar [13,14,15,16,17]. CT scans and X-ray images provide high precision and resolutions, but they are expensive and expose the patient to harmful radiation. Ultrasound technology is noninvasive and harmless but has limited sensitivity. Changes in chest displacement can be observed using depth cameras and radar. However, these methods are constrained by a person’s movement and position making it difficult to observe the displacement accurately. To overcome these issues, various wearable medical devices are being developed that can monitor patients in real time [18]. Among these, near-infrared spectroscopy (NIRS) is gaining attention for its simplicity, low cost, and ability to noninvasively monitor tissue hemodynamics [19].

Common symptoms experienced by patients with respiratory infectious diseases include difficulties in breathing and changes in respiration rate, which are either rapid and shallow (tachypnea) or slow and deep [2,4,20,21]. Therefore, the respiration rate is an important parameter to be monitored [20,21,22,23,24], and various sensors and devices have been developed to measure human respiration rates and patterns. In particular, flexible and wearable sensors have garnered significant interest in terms of respiration monitoring [25,26,27]. Many of these sensors also measure the peripheral arterial blood oxygen saturation (SpO_2_), which has been used as a key marker for the monitoring of the progression of respiratory diseases [28]. Using these sensors to monitor SpO_2_ can be convenient but may result in inaccurate readings due to factors like body movement, reduced blood circulation, the skin thickness, and the skin color [29]. While the SpO_2_ levels are a measure of the systemic arterial blood oxygen saturation, NIRS devices can measure the more localized tissue oxygen saturation (StO_2_) within the microvasculature of tissue [30], which represents an advantage of NIRS devices over pulse oximeters [31]. Studies have shown that NIRS devices can detect the effects of hypoxia better than standard pulse oximeters [32]. Therefore, in this study, we use an NIRS device to evaluate the hemodynamic responses and monitor the respiratory patterns in patients with infectious respiratory diseases.

Infectious respiratory diseases are often transmitted through droplets, person-to-person contact, or vectors [3,33,34,35]. Therefore, diagnosis through direct contact with infected individuals increases the risk of infection for both contacts and healthcare workers [36]. To minimize the occurrence and prevent the spread of highly contagious respiratory diseases from infected individuals to those around them, it is crucial to detect infections early, enforce self-isolation, and ensure appropriate treatment for a swift recovery. To achieve this, non-contact monitoring methods that can check the patient’s condition in real time without direct contact are necessary [37,38,39]. An Internet of Things (IoT)-based monitoring system serves as a prime example. Shafi et al. proposed a system utilizing wearable devices to measure a patient’s temperature, heart rate, and oxygen saturation, transmitting and storing these data in an IoT cloud. Medical staff can access the cloud data to monitor the patient’s health status without direct contact [40]. While this system effectively monitors patients’ conditions without direct contact, clinical decisions must be made by medical staff. However, medical decision-making poses challenges, as healthcare providers face time and physical limitations in diagnosing extensive patient data. Therefore, an intelligent method for the automatic detection of abnormalities from monitored signals is needed.

Deep learning and machine learning algorithms have been applied to various medical fields and clinical decisions, including DNA classification, brain tumor diagnosis, eye diseases, and malaria diagnosis [41,42]. Commonly used algorithms include support vector machines (SVM), extreme gradient boosting (XGBoost), artificial neural networks, and convolutional neural networks (CNN) [43]. For example, Cho et al. utilized a thermal camera to monitor breathing patterns by analyzing temperature fluctuations around the nose regions of individuals. Subsequently, a CNN-based algorithm was employed to automatically categorize psychological stress levels [11]. Shah et al. introduced a technique to detect distinctive patterns associated with COVID-19 from CT scan images utilizing a deep learning algorithm [44]. Hong et al. proposed a method that utilizes mm-wave sensors to measure the displacement of the patient’s chest height and extracts a feature vector using CNN algorithms. The extracted feature vector is compared with pre-learned feature vectors stored in a database to classify it according to the most similar feature vector [45]. Kim et al. developed an embroidered textile sensor that measures electrical impedance changes depending on the intensity of compression and chest movement. The measured electrical impedance changes were classified into respiratory patterns using a 1D-CNN. [46]. Conway et al. used capnography to determine ‘breath’ versus ‘no breath’ using a CNN based on changes in the CO_2_ concentration in a patient’s breathing [47]. These studies have shown encouraging results in classifying individuals suffering from respiratory diseases. However, long-term data collection might cause discomfort to patients, making it unsuitable for continuous monitoring.

In our previous study, we used an NIRS device to measure chest tissue hemodynamic responses and classified simulated breathing patterns using the oxyhemoglobin (O_2_Hb) signal [48]. The simulated breathing patterns included rest, rapid/shallow, and loaded breathing. From the O_2_Hb signals, features including the amplitude, depth, and interval were extracted, and the breathing patterns were classified using the random forest classification method, achieving accuracy of 87%. However, this method is unsuitable for real-time monitoring due to the need for post-processing and feature extraction. Therefore, in another study, we developed a deep learning algorithm based on a 1D-CNN to classify the three breathing patterns, which resulted in classification accuracy of 91.28% [49]. The 1D-CNN model is composed of a single-stream structure. Hence, it faces challenges in parameter learning and optimization as the number of layers increases. As a result, its performance may not improve or may even deteriorate beyond a certain number of layer parameters [50,51,52].

To overcome the issues associated with a single-stream CNN (SCNN), this study proposes a two-stream CNN (TCNN) for breathing pattern classification. The TCNN consists of an autoencoder for the generation of deep compressed feature maps from breathing signals and a classifier module for classification. The deep compressed feature maps are concatenated with feature maps generated by the CNN-based classifier module to ultimately assist in classifying breathing patterns. We hypothesize that the combination of deep compressed feature maps generated by the autoencoder and feature maps created by the CNN-based classifier module will enhance the classification performance and overcome the parameter learning and optimization issues observed in an SCNN.

## 2. Materials and Methods

### 2.1. NIRS Device

The NIRS device consists of one light source with LEDs emitting light at three wavelengths (730 nm, 800 nm, and 850 nm) and two photodetectors, located at 3 cm and 4 cm from the light source (prototype provided by Hamamatsu Photonics K.K., Shizuoka, Japan) (Figure 1). The light source emits a constant amount of light into the tissue, which undergoes absorption and scattering. The backscattered light is detected by the photodetectors, which is then used to calculate changes in O_2_Hb, deoxyhemoglobin (HHb), and total hemoglobin (THb) at each distance. The absolute tissue oxygen saturation (StO_2_) is also measured. The NIRS device was connected to a mobile phone (Pixel XL) for power supply and data recording. A mobile application on the phone converted the raw data measured by the NIRS device into hemodynamic responses, which included two sets of O_2_Hb, HHb, and THb from the two photodetector separations and one set of StO_2_. The data were collected at a sampling rate of 120 Hz, providing high-resolution data.

### 2.2. Breathing Class Definition and Data Collection

To facilitate the non-contact monitoring and classification of breathing patterns for the early detection and monitoring of respiratory disease patients, we define 4 classes of breathing patterns. Patients with respiratory diseases exhibit breathing rates that are faster or slower than those of healthy individuals [2,4,20,21]. In this study, we defined the following four breathing patterns.

Class 1―normal breathing: Participants breathed normally through their mouth or nose while in a relaxed position (Figure 2a).Class 2―breath holding: Participants exhaled and held their breath for as long as they could (Figure 2b).Class 3―slow breathing: Participants breathed at a rate of 10 times per minute for two minutes (Figure 2c).Class 4―rapid breathing: Participants breathed at a rate of 30 times per minute for two minutes (Figure 2d).

The experiment for data collection was conducted at an outpatient clinic at the National Institutes of Health (NIH) Clinical Center. During data collection, the NIRS device was attached to the participants’ chests using medical-grade, double-layer tape. To acquire data on normal breathing, each participant comfortably sat in a chair in a resting position and breathed normally through their mouth or nose for 5 min. The participants then performed a breath holding task where they were asked to exhale and hold their breath for as long as possible, up to two minutes. After resting for two minutes, the task was repeated twice for a total of three repetitions. The participants then rested for five minutes before starting the slow breathing task. During slow breathing, participants took ten breaths per minute for two minutes, rested for two minutes, and then repeated. The slow breathing task ended with a two-minute rest followed by the rapid breathing task, where the participants breathed thirty times per minute for two minutes. This task was also repeated following a two-minute rest period. Figure 2 shows the breathing signals for each class obtained through the NIRS device.

### 2.3. Participants

Twenty-six healthy adult participants (mean age = 34.7 ± 15.7 years old, 18 females) were enrolled in the study. The study enrollment and procedure followed the protocol of a clinical trial and was approved by the Institutional Review Board of the Eunice Kennedy Shriver National Institute of Child Health and Human Development (NICHD protocol #21CH0028) on 31 August 2021. Data from 12 participants were excluded due to motion artifacts and/or low signal-to-noise ratios in one or more tasks. Data from 14 participants who had good signal-to-noise ratios for all four tasks were retained for further analysis.

### 2.4. Data Preprocessing

The data recorded by the NIRS device included impulse noise caused by voltage transients during measurement. To remove the impulse noise, the first derivative of the signal was calculated, and the mean and standard deviation (STD) of the first derivative values were computed. Any signals exceeding the 90% confidence interval were considered noise and removed. Figure 3 shows the signal acquired by the NIRS device (a) and the signal after impulse noise removal (b).

### 2.5. Breathing Pattern Classification Model

The proposed TCNN model consists of a CNN-based autoencoder for the generation of deep compressed features and a CNN-based classifier module for the classification of breathing patterns. The autoencoder comprises an encoder that compresses the input signal and a decoder that reconstructs the high-dimensional features back into the original signal. The high-dimensional features generated from the encoder are trained to detect the most important features that make up the signal. Therefore, we assume that the features generated by the encoder can be helpful in classifying breathing patterns. To test this hypothesis, we designed a breathing pattern classification model, as shown in Figure 4a. The deep compressed features generated by the autoencoder are integrated into the feature model created in the classifier module to classify breathing patterns.

The autoencoder and classifier module were designed by modifying the pre-activation residual model (Pre-ResNet), which is an algorithm developed for the classification of 2D images [53]. Typically, the parameters in a deep learning model are optimized by computing the gradient through the differentiation of the error between the predicted values from the model and the actual values and then performing the backpropagation of this gradient [54,55]. However, as the depth of the model increases, the vanishing gradient problem occurs because the transmitted gradient values decrease, making training difficult and leading to overfitting [50,51,52]. To address these issues, the Pre-ResNet model introduces residual learning by structuring layers in building block units and adding shortcut connections between blocks [53]. Deep learning models composed of residual units mitigate gradient loss during backpropagation by skipping one or more layers. Figure 4b shows a residual unit, which consists of a convolution layer, an activation function (ReLU) [56], and batch normalization (BN) [57].

To apply the residual unit to the 1D signals acquired from the NIRS device, 1D convolutional layers were used instead of 2D convolutional layers, and the size of the convolutional kernels was adjusted to fit the input signal. Detailed information on the network structure and parameters is presented in Table 1 and Table 2, with the BN and ReLU functions omitted. Each convolution block within the same group has the same kernel size and number of kernels. The block structure indicates the kernel size and number of kernels included in each residual block. The block number indicates the number of residual units composing each group. The input size and output size represent the sizes of the input and output features at each stage. The feature map generated in Stage 3 of the encoder is combined with the Stage 3 output features of the classifier module and used to train the classification model.

Stride values of 4 and 3 were used to downsample the feature maps in Stage 0 and Stage 1 of the classifier module and the encoder, respectively. In the remaining stages, except in Stage 2, a stride value of 2 was used in the first convolutional layer of the first residual unit to downsample the feature maps to half their size, and the dimensions of the feature map were expanded twice. As the depth of the network increased, more complex and comprehensive features were extracted.

In the decoder, transposed convolution was used in the first convolutional layer of each stage’s first residual unit to upsample the feature maps to twice their size and halve the dimensionality of the feature maps [58,59]. Stride values of 3 and 4 were used in Stage 2 and Stage 3 to increase the size of the feature maps threefold and fourfold, respectively. As a result, the generated feature maps resembled the input signal more closely as the depth of the network increased.

The total objective function of the TCNN is defined as
(1)$$L_{total}=L_{R}+\alpha L_{C}$$
where $L_{R}$ and $L_{C}$ represent the reconstruction loss of the autoencoder and the cross-entropy loss of the classifier module, respectively. $L_{R}$ is defined as follows:(2)$$L_{R}=\frac{1}{N}\sum_{i=1}^{N}{\left(X_{i}-{\hat{X}}_{i}\right)}^{2}$$
where $X_{i}$ and ${\hat{X}}_{i}$ represent the input signal and the reconstructed signal, respectively, and N is the number of samples.

### 2.6. Dataset

Among the acquired signals, the changes in O_2_Hb and HHb were used to classify breathing patterns. THb, which is the sum of O_2_Hb and HHb, and StO_2_, which represents the relationship between O_2_Hb and THb, were excluded because they are directly dependent on changes in O_2_Hb and HHb. Signals were cropped into intervals of 384 data points, corresponding to approximately 3.2 s of measurement time. As a result, the dataset acquired from the study consisted of a total of 113,456 samples, with 28,364 samples per class. Eleven out of 14 participants were randomly selected to have their data (22,286 samples per class) used for the training of the classification model, while the dataset obtained from the remaining 3 participants (6078 samples per class) was used for model testing.

### 2.7. Data Augmentation

The performance of deep-learning-based algorithms heavily relies on having an extensive training dataset to prevent overfitting. To enhance the size and quality of our training dataset, we applied data augmentation. The main purpose of data augmentation is to increase the quantity and diversity of the training data, and this technique is widely used in various fields, such as computer vision, image classification, and natural language processing [60,61,62,63]. To determine the data augmentation methods, we observed changes in the data during each participant’s breathing. The participants exhibited similar respiratory rates while performing respiration corresponding to each class. On the other hand, the amplitude, shape, and direct current (DC) offset of the signals varied among the participants. Hence, a random amplitude scale, horizontal flipping with 50% probability, and DC offset were applied as data augmentation methods to maintain the signal period and increase the diversity of the signal amplitude, shape, and DC values. Figure 5 shows an example of data augmentation.

Each data augmentation method was applied before the signal was input into the model and is defined as follows.

DC offset: performed by adding a DC value β to the original signal.(3)$${f_{DC}}_{offset}\left(I_{s}\right)=I_{s}+\beta$$
where I represents the sample data input to the model during model training, and the subscript s denotes the data source. Let D and d represent the training dataset and the data sample belonging to the training dataset, respectively; then, the range of β is defined as follows.
(4)$$\beta \in \left[\underset{d_{s}^{i}\in D_{s}}{\min}\left(\min\left(d_{s}^{i}\right)\right)-\min\left(I_{s}\right),\underset{d_{s}^{j}\in D_{s}}{\;\max}\left(\max\left(d_{s}^{j}\right)\right)-\max\left(I_{s}\right)\right]$$
where i,j∈[1,2,…,N] represent the indices of the data samples, and N denotes the total number of samples in the training dataset.


Amplitude random scaling: involves reducing or increasing the amplitude of the data by multiplying the scaling value γ with the original data.

(5)
$$f_{Scale}\left(I_{s}\right)=I_{s}\times \gamma$$



γ is determined by the smallest and largest amplitude values among the training dataset.
(6)$$\gamma \in \left[\underset{d_{s}^{i}\in D_{s}}{\min}\left(\max\left(d_{s}^{i}\right)-\min\left(d_{s}^{i}\right)\right),\;\underset{d_{s}^{j}\in D_{s}}{\max}\left(\max\left(d_{s}^{j}\right)-\min\left(d_{s}^{j}\right)\right)\right]$$
Horizontal flip $f_{Flip}\left(I_{s}\right)$: performed by flipping the original data horizontally and occurs with a probability of 50%. Horizontal flipping is a simple technique that can effectively improve a model’s learning performance.

In this experiment, the augmented data $f_{Aug}\left(I_{s}\right)$ are finally defined as follows:(7)$$f_{Aug}\left(I_{s}\right)=f_{Flip}\left(f_{DC\;offset}\left(f_{Scale}\left(I_{s}\right)\right)\right)$$

## 3. Results

### 3.1. TCNN with and without Autoencoder

We used a computer equipped with a CPU i5-12600k, a GPU RTX-3090, and 64 GB DDR4 RAM for the experiment. We first pre-trained an autoencoder with the same structure as in Figure 4. The weights of the pre-trained autoencoder were used as the initial values of the autoencoder of the TCNN. The pre-training of the autoencoder and the training of the TCNN were conducted using stochastic gradient descent (SGD) for 140 epochs, with momentum of 0.9 and a batch size of 128. The initial learning rate for the pre-training of the autoencoder was set to 0.1, while the initial learning rate for the training of the TCNN was set to 0.01. The learning rate was divided by 10 every 30 epochs. In this experiment, the hyper-parameter α was set to 10.

Table 3 presents the experimental results to assess the impact of the compressed feature maps generated from the autoencoder on the classification performance. Each model was tested five times. The SCNN (without an autoencoder) is a model derived from the TCNN excluding the autoencoder. The SCNN has no concatenation group, and the output size of Stage 3 is 1 × 32. The classification accuracy is defined by the following formula.
(8)$$Accuracy\left(\%\right)=\frac{\mathrm{Number}\;\mathrm{of}\;\mathrm{correct}\;\mathrm{predictions}}{\mathrm{Total}\;\mathrm{number}\;\mathrm{of}\;\mathrm{predictions}}\times 100\left(\%\right).$$

The experimental results showed that the TCNN models had better performance than the SCNN models. In addition, the SCNN with more layers (264 layers) showed slightly worse performance than the SCNN with less layers (201 layers). Furthermore, applying data augmentation techniques during the training process of the TCNN model led to an approximately 5% improvement in the classification accuracy compared to not applying data augmentation.

### 3.2. Confusion Matrix

Figure 6 shows the normalized confusion matrix for the breathing pattern classification results. The ground truth represents the labels for the data samples, while the predicted labels indicate the results predicted by the model. Therefore, a larger value in the diagonal matrix indicates that the predicted label has been correctly classified. It can be observed that the diagonal matrix is prominent in all three confusion matrices. This indicates that the majority of the data samples have been correctly classified into their respective classes. It is worth noting that in all three confusion matrices, the normal class has the lowest classification accuracy.

### 3.3. Performance of Well-Known Classification Algorithms

Table 4 shows a performance comparison of several classification methods. To compare the classification performance of the proposed method with that of other state-of-the-art classification algorithms, we used random forest [64], EfficientNetV2-M [65], PyramidNet [66], a modified 1D-PreResNet [49], and a coarse-to-fine CNN (CF-CNN) [67]. The random forest algorithm consists of multiple decision trees and is an ensemble learning method used for the training and classification of data. For our experiment, we employed a model consisting of 100 decision trees. To train the random forest, we utilized the average amplitude, depth, and interval of the NIRS signals [48]. EfficientNetV2-M and PyramidNet are CNN-based algorithms developed for image classification, renowned for their exceptional performance and stability and widely utilized across various domains [68,69,70,71]. PyramidNet consists of 272 layers, and the widening step factor value for increasing feature map dimensions is set to 200. CF-CNN consists of a main network for fine classification and several coarse sub-networks with a hierarchical structure for coarse classification. To train a new hierarchical structure for coarse classification, we generated hierarchical group labels using the group label augmentation method [67]. The coarse network of CF-CNN consists of a downsized version of the base network that constitutes the main network. This model is also proposed to overcome the limitations of learning in single-stream networks. For comparison, the 2D convolutional layers of each network were replaced with 1D convolutional layers. For the evaluation of the CF-CNN model, we chose PyramidNet as the base network. The number of layers in each of the Coarse 1, Coarse 2, and main networks was set to 26, 52, and 272, respectively. For Coarse 1 and Coarse 2 classification, the number of group labels was set to 2 and 3, respectively.

Each model was tested five times, and, for each test, the model was trained from scratch. To train the 1D Pre-ResNet, EfficientNetV2-M, PyramidNet, and CF-CNN, the input data were downsampled using the bilinear interpolation method to match the input size of each model at the input layer. While our proposed method had a larger number of parameters and floating-point operations (FLOP) compared to the 1D Pre-ResNet, it achieved the highest classification performance among the compared methods.

## 4. Discussion

The aim of our study was to develop a breathing pattern classification method for the real-time monitoring of patients with infectious respiratory diseases such as COVID-19. First, we defined four common breathing patterns observed in patients with respiratory diseases: normal, breath holding, slow, and rapid. Then, we collected data from participants using a wearable NIRS device. Second, we proposed the TCNN, consisting of an autoencoder and a classifier module, to classify breathing patterns. We hypothesized that the deep compressed features generated by the encoder of the autoencoder would contain information about the composition of the signal, which could potentially help to improve the classification performance. To validate this hypothesis, we compared the classification performance of the proposed TCNN with that of an SCNN model. Comparing the breathing pattern classification accuracy between the two SCNN models with 201 layers and 264 layers, we observed that the SCNN with less layers exhibited higher classification accuracy. This demonstrates the challenge of parameter learning becoming more difficult as the number of parameters increases. Comparing the classification performance of the TCNN and SCNN, we observed that the TCNN achieved higher classification performance. This result suggests that the deep compressed features generated by the encoder of the autoencoder contribute to the improvement in classification performance. Furthermore, as training is performed on networks with different objectives, the diversity of the feature maps is enhanced, and the burden on parameter optimization is reduced.

The results of the TCNN (without data augmentation) showed significantly lower classification accuracy compared to the TCNN, primarily due to insufficient training data. Deep learning algorithms can prevent overfitting and enhance the model’s generalization when trained with diverse and abundant data. To improve the model’s generalization performance, we applied three data augmentation methods: DC offset, amplitude random scale, and horizontal flip. As a result, the classification performance was improved by approximately 5%. This outcome suggests that data augmentation effectively enhanced the model while preserving the essential features necessary for breathing pattern classification.

For all three models in Figure 6, the classification accuracy for normal breathing was the lowest. One plausible explanation is that the most distinguishing feature in breathing pattern classification is the respiratory rate. In some instances, a normal breathing rate can be confused with slow and rapid breathing.

Among all classification models used in this study, the proposed TCNN model exhibited the best classification accuracy. Moreover, the TCNN model used significantly a smaller number of parameters and FLOP, which are directly related to the hardware storage space and computational load. FLOP represents the number of floating-point operations required to execute each classification model. Therefore, a lower FLOP count indicates a shorter classification time. Although the TCNN shows slightly more parameters and FLOP than the 1D-PreResNet, which has the fewest parameters and FLOP, it achieves higher classification accuracy. When compared to the performance of the EfficientNetV2-M, PyramidNet, and CF-CNN models, the proposed method demonstrates the best performance with significantly fewer parameters and FLOP. The classification accuracy when using the random forest algorithm was the lowest among the compared methods. Traditional machine learning methods like random forest perform well when the input features are diverse and contain distinct information. However, creating diverse feature extractors manually is very challenging for humans. On the other hand, deep learning algorithms show better performance than traditional machine learning methods because they automatically generate diverse and meaningful features during the model learning process.

Future studies could expand on our results by collecting signals from individuals and patients with respiratory diseases. We are considering the application of noise removal techniques during the data preprocessing stage to enhance the classification performance. Additionally, we are exploring the integration of sensors such as temperature sensors, accelerometers, and NIRS devices to obtain diverse and meaningful signals.

## 5. Conclusions

This study proposed a two-stream convolutional neural network (TCNN) model for the classification of breathing patterns to facilitate the evaluation and monitoring of patients with infectious respiratory diseases. This method utilizes the changes in oxyhemoglobin and deoxyhemoglobin measured using a wearable NIRS device. The proposed TCNN consists of an autoencoder and a classifier module. The deep compressed feature maps generated in the encoder of the autoencoder are combined with the feature maps generated in the classifier module to classify breathing patterns. The compressed features generated by the encoder contain important information that constitutes the signal, unlike the features generated by the classifier. Therefore, compressed features help to improve the breathing pattern classification performance because they provide additional information for the classification of breathing patterns. The proposed method alleviates the issue of decreasing parameter learning capabilities as the depth of a typical single-stream network model increases. For the breathing pattern classification experiment, we defined four common breathing patterns observed in infectious respiratory disease patients: normal breathing, breath holding, slow breathing, and rapid breathing. We then conducted the classification experiment. As a result, we achieved classification accuracy of 94.63%. The proposed method demonstrates its potential for the remote and real-time monitoring of respiratory diseases such as COVID-19, pneumonia, and influenza.

## Figures and Tables

**Figure 1 bioengineering-11-00709-f001:**
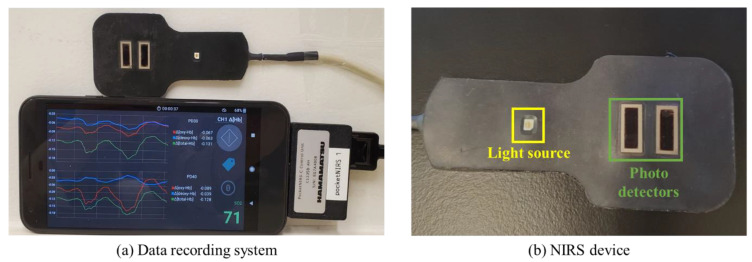
Data recording system and NIRS device. (**a**) The signals measured by the NIRS device are converted into O_2_Hb, HHb, THb, and StO_2_ in real time and recorded on the connected mobile phone. (**b**) The NIRS device consists of one light source and two photodetectors.

**Figure 2 bioengineering-11-00709-f002:**
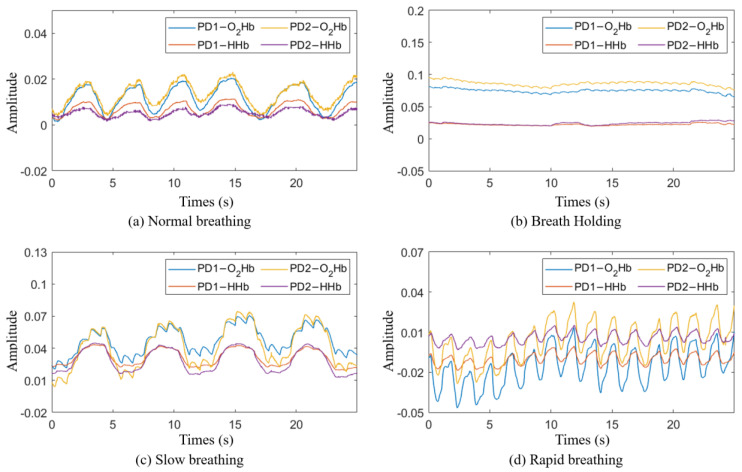
Breathing signal in the time domain: (**a**) normal breathing; (**b**) breath holding; (**c**) slow breathing; (**d**) rapid breathing.

**Figure 3 bioengineering-11-00709-f003:**
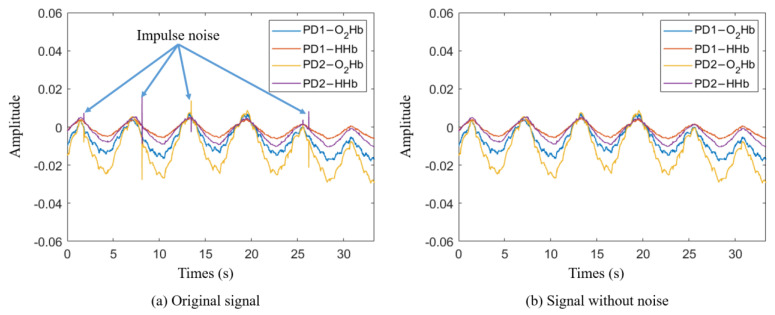
Impulse noise removal: (**a**) original signal from NIRS device, (**b**) signal with impulse noise removed.

**Figure 4 bioengineering-11-00709-f004:**
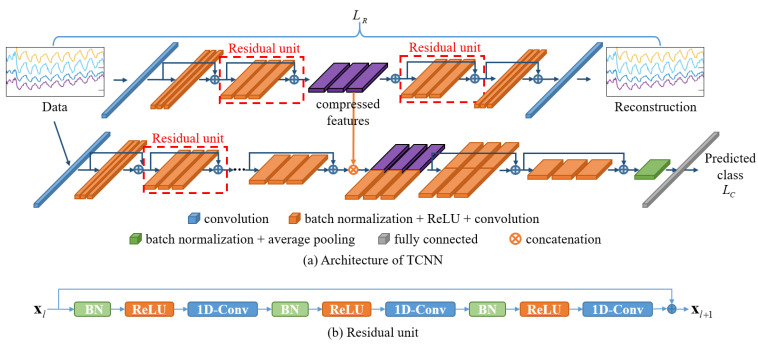
Architecture of TCNN. (**a**) The autoencoder network is used to generate compressed features from the input data; then, the compressed features are combined with deep features extracted from the classification network. (**b**) The structure of a residual unit.

**Figure 5 bioengineering-11-00709-f005:**
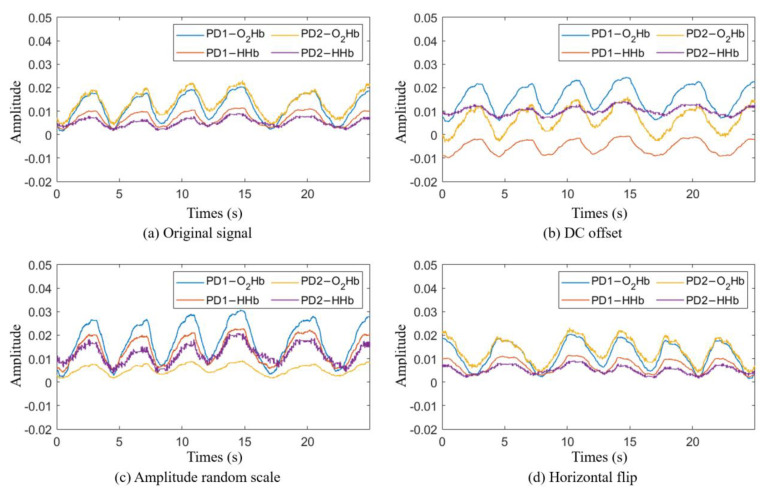
Example of the data augmentation methods. (**a**) The original signals recorded using the mobile phone. (**b**) The result of adding DC offset values of 0.004, −0.012, −0.007, and 0.005 to the PD1-O_2_Hb, PD1-HHb, PD2-O_2_Hb, and PD2-HHb signals, respectively. (**c**) The result of multiplying the PD1-O_2_Hb, PD1-HHb, PD2-O_2_Hb, and PD2-HHb signals by scale values of 1.5, 2, 0.4, and 2.3, respectively. (**d**) The result of a horizontal flip occurring with a 50% probability.

**Figure 6 bioengineering-11-00709-f006:**
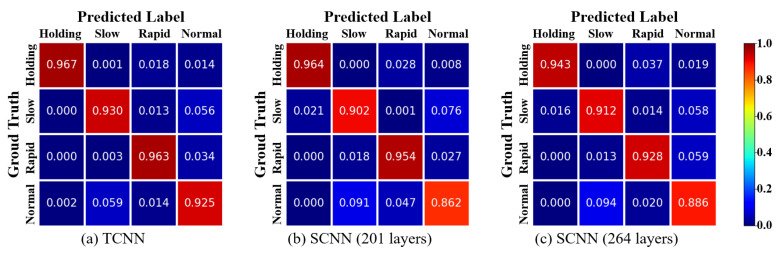
Confusion matrix from predicted results of the TCNN and SCNN (without autoencoder).

**Table 1 bioengineering-11-00709-t001:** Autoencoder architectures.

	Group Name	Input Size Output Size	Block Structure (Kernel Size, Number)	Block Number
Encoder	Stage 0	1 × 384 1 × 96	1 × 7, 16	1
	Stage 1	1 × 96 1 × 32	1 × 7, 16	1
	Stage 2	1 × 32 1 × 32	1 × 1, 16 1 × 3, 16 1 × 1, 64	5
	Stage 3	1 × 32 1 × 16	1 × 1, 32 1 × 3, 32 1 × 1, 128	5
Decoder	Stage 0	1 × 16 1 × 32	1 × 1, 128 1 × 3, 32 1 × 1, 32	5
	Stage 1	1 × 32 1 × 32	1 × 1, 64 1 × 3, 16 1 × 1, 16	5
	Stage 2	1 × 32 1 × 96	1 × 5, 16	1
	Stage 3	1 × 96 1 × 384	1 × 7, 4	1

**Table 2 bioengineering-11-00709-t002:** Classifier module architectures.

Group Name	Input Size Output Size	Block Structure (Kernel Size, Number)	Block Number
Stage 0	1 × 384 1 × 96	1 × 7, 16	1
Stage 1	1 × 96 1 × 32	1 × 7, 16	1
Stage 2	1 × 32 1 × 32	1 × 1, 16 1 × 3, 16 1 × 1, 64	22
Stage 3	1 × 32 1 × 16	1 × 1, 32 1 × 3, 32 1 × 1, 128	11
Concatenation	1 × 16 → 1 × 32	-	-
Stage 4	1 × 32 1 × 16	1 × 1, 32 1 × 3, 32 1 × 1, 128	11
Stage 5	1 × 16 1 × 8	1 × 1, 64 1 × 3, 64 1 × 1, 256	22
Average pooling	1 × 8 1 × 256	1 × 8	1
Fully connected layer	1 × 256 Class number	-	1

**Table 3 bioengineering-11-00709-t003:** Classification performance compared with an SCNN (without autoencoder) and the data augmentation effect.

Method	Mean Accuracy (%)	STD	Best Accuracy (%)
TCNN (201 layers)	94.00	0.449	94.63
TCNN (without data augmentation) (201 layers)	88.51	0.337	89.04
SCNN (201 layers)	91.50	0.458	92.04
SCNN (264 layers)	91.13	0.443	91.74

**Table 4 bioengineering-11-00709-t004:** Classification performance compared with state-of-the-art methods.

Method	Number of Parameters	FLOP (for 1 Sample)	Mean Accuracy (%)	STD	Best Accuracy (%)
Random Forest [64]	-	-	88.10	0.321	88.49
1D Pre-ResNet (200 layers) [49]	1.36 M	14.8 M	90.98	0.428	91.54
EfficientNetV2-M [65]	52 M	61.6 M	93.96	0.459	94.45
PyramidNet [66]	17 M	198 M	89.74	0.412	90.17
CF-CNN [67]	29.7 M	244 M	91.51	0.406	92.01
Proposed Method	1.53 M	16.4 M	94.00	0.449	94.63

## Data Availability

Data supporting the reported results available at https://github.com/dkskzmffps/TCNNforBP, accessed on 11 July 2024.

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
