# Peer review of "Two-Stream Convolutional Neural Networks for Breathing Pattern Classification: Real-Time Monitoring of Respiratory Disease Patients"

_bioengineering, 2024, doi:10.3390/bioengineering11070709_

Round 1
Reviewer 1 Report
Comments and Suggestions for Authors
The paper addresses an important issue of classifying breathing patterns by the patients having respiratory issues. There is a paper at https://doi.org/10.3390/s23125592 with same authors, and a similar title, on which authors need to comment as to where and how they are different in the new submission. Additionally, I request the authors to respond to following queries:
1. Why a two-stream convolutional neural network (TCNN) for breathing pattern classification has been selected and no other ML approach has been tested for this purpose?
2. The continuous monitoring of patients with infectious respiratory diseases is attempted but not much details on this are provided within the manuscript.
3. The 14 participants have been used by the authors to collect the data, is this sample size enough for the correct identification of the problem? What could be its margin of error and confidence level once results are achieved with a low sample size?
4. Why the hemodynamic responses are measured using only a single device and calling it our device is to be clarified.
5. Though authors claim to identify 22 respiratory changes but yet only work on 4 such cases.
6. Many latest related research works must be made part of the Introduction and Literature review section, few are given below
https://doi.org/10.3390/s23115275
https://doi.org/10.1371/journal.pone.0298582
https://doi.org/10.3390/s23125736
https://doi.org/10.1007/s10877-023-01028-y
https://doi.org/10.1002/cpe.6342
Reviewer 2 Report
Comments and Suggestions for Authors
Regarding the paper "Two-Stream Convolutional Neural Networks for Breathing Pattern Classification: Real-Time Monitoring of Respiratory Diseases Patients," I would like to communicate that, although the methodology is valuable and deemed appropriate, there are a few small edits that need to be made. After implementing these simple adjustments, the manuscript can be recommended for publication.
1. Because not all findings are sufficiently illustrated, the abstract lacks comprehensive development and maturity. It is recommended that the abstract contain all of the findings.
2. To keep readers interested and improve the manuscript's readability, it is advised to go into greater detail in the introduction about the uses of machine learning in clinical data analysis. In order to do this, citing relevant studies such
-
"AI-driven malaria diagnosis: developing a robust model for accurate detection and classification of malaria parasites" and "Toward artificial intelligence (AI) applications in the determination of COVID-19 infection severity: considering AI as a disease control strategy in future pandemics" would not only showcase additional applications of artificial intelligence in data analysis but also enrich the manuscript's reference list and improve the introduction's readability.
-
Optionally, consider including a "suggestions for the future" section to broaden the perspectives of future researchers.
By incorporating these revisions, the study's quality will be enhanced, elevating its scientific value to a level suitable for publication.
Reviewer 3 Report
Comments and Suggestions for Authors
Please find the attached review report.

Reviewer 4 Report
Comments and Suggestions for Authors
The Abstract is good - very informative, however too technically detailed - try to generalize it a little - just recommendation
The Introduction is very good - outlining the motivation behind the research, referring a solid set of research, addressing the authors' previous work.
Methods section and results section are structured well and provide full overview of the experiment, all stages are presented clearly and coherently. Results are described in a consistent way.
Discussion is ok too.
I would recommend to add 2 sections - separate literature overview and limitation sections. If the authors make conclusion a bit more detailed with bullet-points of what exactly are real-world potential application of the achieved results (by the way - there are some repetitions all over the text on the same things - please check). Please provide some future work plans too.
there are also some spelling things - like matrice - for matrix, or grammar things - like sensitive for sensitivity - please check. Overall style is ok, easy to read and follow.
Comments on the Quality of English Language
Minor editing
Round 2
Reviewer 1 Report
Comments and Suggestions for Authors
My comments are addressed.